# Proximate Drivers of Population-Level Lizard Gut Microbial Diversity: Impacts of Diet, Insularity, and Local Environment

**DOI:** 10.3390/microorganisms10081550

**Published:** 2022-07-31

**Authors:** Virginie Lemieux-Labonté, Chloé Vigliotti, Zoran Tadic, Beck Wehrle, Philippe Lopez, Eric Bapteste, François-Joseph Lapointe, Donovan P. German, Anthony Herrel

**Affiliations:** 1Département de Sciences Biologiques, Université de Montréal, C.P. 6128, Succ. Centre-Ville, Montréal, QC H3C3J7, Canada; virginie.lemieux-labonte@umontreal.ca (V.L.-L.); francois-joseph.lapointe@umontreal.ca (F.-J.L.); 2UMR 518, MIA Paris-Saclay, Département de Mathématiques et Informatique Appliquées, AgroParisTech, 16 rue Claude Bernard, 75005 Paris, France; 3Department of Biology, Faculty of Science, University of Zagreb, Rooseveltov trg 6, 10000 Zagreb, Croatia; zoran.tadic@biol.pmf.hr; 4Department of Ecology and Evolutionary Biology, 5309 McGaugh Hall (Lab), University of California, Irvine, CA 92697, USA; beck.wehrle@gmail.com (B.W.); dgerman@uci.edu (D.P.G.); 5Institut de Systématique, Évolution, Biodiversité (ISYEB), Muséum National d’Histoire Naturelle, CNRS, Sorbonne Université, Ecole Pratique des Hautes Etudes, Université des Antilles, CP 50, 57 rue Cuvier, 75005 Paris, France; philippe.lopez@upmc.fr (P.L.); epbapteste@gmail.com (E.B.); 6UMR 7179, CNRS/MNHN, Département d’Écologie et de Gestion de la Biodiversité, CEDEX, 75005 Paris, France; anthony.herrel@mnhn.fr

**Keywords:** gut microbiota, lizard, omnivory, 16S rRNA gene, insularity

## Abstract

Diet has been suggested to be an important driver of variation in microbiota composition in mammals. However, whether this is a more general phenomenon and how fast changes in gut microbiota occur with changes in diet remains poorly understood. Forty-nine years ago, ten lizards of the species *Podarcis siculus* were taken from the island of Pod Kopište and introduced onto the island of Pod Mrčaru (Croatia). The introduced population underwent a significant dietary shift, and their descendants became omnivorous (consuming up to 80% plant material during summer). Variation in their gut microbiota has never been investigated. To elucidate the possible impact on the gut microbiota of this rapid change in diet, we compared the microbiota (V4 region of the 16S rRNA gene) of *P. siculus* from Pod Mrčaru, Pod Kopište, and the mainland. In addition, we explored other drivers of variation in gut microbiota including insularity, the population of origin, and the year of sampling. Alpha-diversity analyses showed that the microbial diversity of omnivorous lizards was higher than the microbial diversity of insectivorous lizards. Moreover, omnivorous individuals harbored significantly more *Methanobrevibacter.* The gut microbial diversity of insectivorous lizards was nonetheless more heterogeneous. Insectivorous lizards on the mainland had different gut microbial communities than their counterparts on the island of Pod Kopište. *Bacillus* and *Desulfovibrio* were more abundant in the gut microbiota from insular lizards compared to mainland lizards. Finally, we showed that the population of origin was also an important driver of the composition of the gut microbiota. The dietary shift that occurred in the introduced population of *P. siculus* has had a detectable impact on the gut microbiota, but other factors such as insularity and the population of origin also contributed to differences in the gut microbial composition of these lizards, illustrating the multifactorial nature of the drivers of variation in gut microbiota. Overall, our data show that changes in gut microbiota may take place on ecological timescales. Yet, diet is only one of many factors driving variation in gut microbiota across populations.

## 1. Introduction

Diet is known to affect the physiology and morphology of many taxa [1]), including the composition of the gut microbiota [2,3,4]. We use the term ‘microbiota’ to refer to the taxonomic diversity of Bacteria and Archaea assessed using marker genes, rather than ‘microbiome’, which refers to both taxonomic and functional diversity of the complete community [5]. The microbial taxa hosted in the gut change during the development and aging of animals [6], including humans [7,8,9,10]. The microbiota also changes over longer time scales in relation to diet [2]. A correlation between host phylogeny and enteric microbiome composition and abundance [11,12] has been observed, suggesting coevolution of host and microbiome at deeper time scales. In addition to dietary and genetic variation, many other factors, such as geographical and physicochemical characteristics of the environment, are known to impact the gut microbiota (e.g., [13,14,15]). Consequently, how fast gut microbiota can change and how strong the impact of diet is versus other environmental features in driving variation in gut microbiota remains relatively poorly understood.

To address how diet impacts the enteric microbiota, we investigated a unique system featuring rapid changes in diet at the population level. Forty-nine years ago, Nevo and colleagues (1972) designed a study to analyze the competition between *Podarcis siculus* and *Podarcis melisellensis* on islands [16]. They introduced ten *P. siculus* from the island of Pod Kopište to the island of Pod Mrčaru, and ten *P. melisellensis* from Pod Mrčaru to Pod Kopište. During a follow-up study, it was noted that *P. melisellensis* has disappeared from Pod Mrčaru, while the descendants of the ten *P. siculus* introduced on this island have thrived [17]. Moreover, the Pod Mrčaru lizards are currently largely omnivorous (consuming up to 80% plant matter in the diet in summer) and have undergone significant morphological changes in the hindgut, including the development of caecal valves [17]. These changes are typically associated with the consumption of plant material in lizards (see [18] for an overview). There are also subtle changes in digestive biochemistry between the lizards on the two islands, and these changes are largely isolated to the hindgut [19], which houses the enteric microbiota engaged in the digestive process [6,20]. Given the known impact of diet on the gut microbiota in vertebrates and the fact that vertebrates are unable to endogenously digest plant fiber (e.g., cellulose; [1]), we predicted there would be differences in the gut microbiota among the *P. siculus* lizards from the two islands.

Previous studies on the highly specialized herbivorous marine and land iguanas of the Galápagos Islands showed that these animals had different microbiota from mammalian herbivores [21,22]. Interestingly, both species showed the conservation of microbial genes important to the breakdown process of plant material [21,22]. However, how general these changes are and how fast changes in the microbiota evolve remains unclear (e.g., the Galápagos iguanas diverged 4.5 million years ago; [23]. A study experimentally manipulating the diet of an omnivorous species of *Liolaemus* lizards showed that an experimental increase in the amount of plant matter in the diet over 40 days significantly impacted the gut microbiota, suggesting that some changes may happen rapidly [20]. Kohl et al. (2016) further observed that lizards fed a plant-only diet had a higher gut microbial diversity than lizards fed a mixed diet [20]. However, other studies have suggested that lizard gut microbiomes are at least partly derived from the local environment [6,24,25]. For example, recent studies have demonstrated that altitude, geography, and insularity impacted the gut microbial diversity in lizards [25,26,27]. Overall, these studies suggest that in lizards, diet is not the only driver of variation in the gut microbiota, but also that the local environment may play a crucial role in establishing its composition.

To explore the impact of the relatively rapid changes in diet as well as the influence of the local environment (population of origin) and sampling year (2014 vs. 2016) in natural populations of lizards, we used 16S rRNA gene sequencing to compare the gut microbial communities of 40 *P. siculus* sampled from the islands of Pod Kopište and Pod Mrčaru, and an insectivorous population from the mainland (Split). Thus, we compared recently derived populations from the islands and contrasted them with a more distantly related population of *P. siculus*. We also compared the three populations across different years (2014 and 2016) to examine whether any differences among the populations were stable over time. We hypothesized that the lizards from Pod Mrčaru would have more microbial taxa associated with the digestion and metabolism of plant material [6,21] than the insectivorous populations, and that the island lizards would be different from the lizards on the mainland [25]. The overall aim of the study was to better understand the possible drivers of gut microbial diversity in lizards with different diets (omnivorous vs. insectivorous) and from different localities (insular vs. mainland).

## 2. Material & Methods

### 2.1. Sample Collection

Forty *P. siculus* were collected in 2014 and in 2016 (twenty-two in 2014, and eighteen in 2016), yielding twelve insectivorous lizards from Pod Kopište (seven males and five females), thirteen omnivorous lizards from Pod Mrčaru (eight males and five females), and fifteen lizards from the mainland (ten males and five females; Figure 1, Appendix A). Previous studies based on stomach contents analysis show that lizards from the population from Split consume on average 17.7% plant matter by volume [28]. The lizards from the island of Pod Kopište consume on average between 6.5 and 50% plant matter by volume depending on the season and the year. By contrast, the lizards of the population on Pod Mrčaru include between 54 and 78% plant matter by volume into their diet depending on the season and year of sampling [17,28]. The insular populations are composed of small islets ranging between 2931 m^2^ for Pod Mrčaru to 7915 m^2^ for Pod Kopište. Although population density has never been quantified using formal capture-mark-recapture methods, estimates based on transects range from 3082 lizards per hectare on Pod Mrčaru to 1045 lizards per hectare on Pod Kopište [29]. Although we did not find data for *P. siculus* from the mainland of Croatia, data for a population in central Italy suggest average densities ranging from five to 24 lizards per hectare [30]. Whereas the mainland harbors a complex lizard community [31], the only other lizard on the two islets is the insectivorous lacertid lizard, *D. oxycephala* ([32]; A.H., personal observation). However, this species is a specialist rock dweller and does not inhabit the vegetated areas of the islets. Finally, the two islands are separated by deep water and both islands are far from the mainland, making admixture between populations unlikely. Ongoing genetic studies indeed suggest no evidence of admixture (Anamaria Štambuk, personal communication).

Animals were collected and euthanized in the field using an intramuscular injection of pentobarbital. Gut samples were collected immediately after euthanasia and frozen less than 10 min after animals were euthanized. We dissected the distal intestine immediately after euthanasia and squeezed the contents out onto a chilled RNAse free surface with a flat tool (the back of a razor blade). Each sample was immediately frozen at −80 °C. All procedures were approved by the institutional animal care and use committee and under a permit from the Croatian Ministry of the Environment (permit no. 517-07-1-1-1-16-6).

### 2.2. 16S rRNA Gene Sequencing and Processing

Samples were kept at −80 °C until DNA extraction (less than six months). We used PCR primers 515/806 with barcodes on the forward primer to sequence the 16S rRNA gene V4 variable region in a 30 cycle PCR using the HotStarTaq Plus Master Mix Kit (Qiagen, Germantown, Maryland, USA) under the following conditions: 94 °C for 3 min, followed by 28 cycles of 94 °C for 30 s, 53 °C for 40 s and 72 °C for 1 min, after which a final elongation step at 72 °C for 5 min was performed. After amplification, PCR products were checked in 2% agarose gel to determine the success of amplification and the relative intensity of bands. Multiple samples were pooled together in equal proportions based on their molecular weight and DNA concentrations. Pooled samples were purified using calibrated Ampure XP beads. The pooled and purified PCR product were used to prepare DNA library by following Illumina TruSeq DNA library preparation protocol. Sequencing was performed at MR DNA (www.mrdnalab.com, accessed on 29 November 2016, Shallowater, TX, USA) on a MiSeq using V3 chemistry, following the manufacturer’s guidelines. Sequence data were processed using MR DNA analysis pipeline (MR DNA, Shallowater, TX, USA). In summary, paired end sequences were merged, depleted of barcodes and primers, and sequences <150 bp were removed, as were sequences with ambiguous base calls. The QIIME software package version 1.9.1 was used for subsequent steps [33]. Operational Taxonomic Units (OTUs) were retained at ≥97% similarity (pick_open_reference_otus.py using uclust) against the SILVA database version 128 [34]. Contaminant chloroplast and mitochondria were filtered out prior to subsequent analysis.

We obtained 5,493,157 reads from the V4 region of the 16S rRNA gene across the 40 samples analyzed (Appendix A). These reads were pooled and cleaned to remove chimeras, which retained 87.2% of the reads. These clean reads were clustered in OTUs (≥97%) with QIIME (pick_open_reference.py, keeping only OTUs with more than three reads).

### 2.3. Statistical Analyses

All analyses were performed in R [35].

### 2.4. ANCOM Analysis

At the compositional level, the abundance of microbial taxa at the genus level was compared between groups (i.e., diet, location, year, sex) using an Analysis of Composition of Microbiomes (ANCOM) [36] using the *ancom.R* package. ANCOM is based on nonparametric tests (i.e., either Kruskal–Wallis for independent samples, or Friedman test for dependent samples) and is appropriate for compositional data [37]. The test relies on point estimates of data transformed by an additive log ratio, where presumed invariant taxa are selected as the denominators. To highlight predominant differences between microbiota samples, the analysis was performed using an OTU table only including OTUs with a relative abundance in the entire dataset that was higher than or equal to 1%.

### 2.5. Alpha Diversity

The alpha diversity of the gut microbial community of each sample was computed using the Shannon index [38,39]. The Shannon index, which includes both OTU richness and evenness, was selected due to its reduced sensitivity to sample depth differences ([40,41]; Appendix A). Conventional Shannon index interpretation is limited because of non-linearity issues. Indeed, when species numbers in an equally distributed community double, the Shannon index value will not double [42,43]. An exponential transformation of the Shannon index transforms this in true diversity and allows a straightforward interpretation of the results. True alpha diversity values were compared using linear mixed-effect model (*lmer()* function) and significance was tested with Likelihood Ratio Tests with a chi-square distribution [44]. We calculated the Shannon index on non-rarified data as this measure is robust to variation in sequencing depth (Appendix A).

### 2.6. Beta Diversity Index and Ordination

Three distinct phylogenetic distances, unweighted UniFrac (qualitative), weighted UniFrac (quantitative), and Bray–Curtis [45] were computed on rarefied data (30,881 sequences/sample) as these measures can be sensitive to differences in sequencing depth [46,47]. Rarefaction (with the rarefy *even_depth()* function) and beta diversity computations were performed with the *phyloseq* package [48]. All beta diversity results were visualized with principal coordinates analysis (PCoA; [49]) using the *ordinate()* function. The distance matrix was checked with the *is.euclid()* function of the *ade4* package [50] prior to the ordination to ensure that all distances were Euclidian and properly representable by PCoA [51]. When required, square-root transformations were applied to obtain distance matrices satisfying the Euclidian condition (Weighted UniFrac and Bray–Curtis). To test the homogeneity of dispersion within groups (PERMDISP), we used the *betadisper()* function. The homogeneity of multivariate dispersions within groups is based on the estimation of the deviation from the group centroid [52]. The null hypothesis of this test is that the average within-group dispersion is identical in all groups [53].

To assess the influence of explanatory variables on the microbiota composition, we used distance-based redundancy analysis (db-RDA), a method intended to conduct a redundancy analysis (RDA) on distance matrices [54]. It is computed by first decomposing UniFrac distances (weighted or unweighted) into principal coordinates and then applying RDA to the corresponding principal coordinates using the *capscale()* function of the R package *vegan* [55,56]. To better understand the relationships among explanatory models in the variation of the microbial assemblages, partial db-RDA was also computed [57]. This form of RDA allows for exploration of the contribution of an explanatory variable in the model while controlling for other explanatory models. Adjusted R-squared (*R*^2^) values [58] were calculated to compare the explanatory power of such models containing different numbers of variables. Significance of db-RDA and partial db-RDA was tested via 9999 permutations with the *anova.cca()* function of the R package *vegan* [56].

## 3. Results

### 3.1. Analysis of Gut Microbiota Composition

The analysis of the composition of microbiomes (ANCOM) was performed on unrarefied OTU tables with a relative abundance higher or equal to 1%. Unassigned taxa represented a mean of 0.9% of the gut microbiota (median = 0.3, SD = 2%, range = 0.1–11%). Fourteen OTUs are part of the more abundant microbial organisms in the gut of *P. siculus* (relative abundance ≥ 1%) (Table 1). The Clostridiales order is the most diversified, with some OTUs identified at the ordinal level and others at the family level, including Clostridiaceae, Lachnospiraceae, and Peptostreptococcaceae (Appendix A). OTUs assigned to Enterobacteriacea and to the genus *Citrobacter* were also among the most abundant taxa. Overall, the omnivorous and insectivorous lizards showed slight, but significantly different microbial communities in their guts (Figure 2 and Figure 3). The analysis of composition detected differently abundant taxa at different classification levels (Table 1). The Archaean, *Methanobrevibacter*, was more abundant in omnivorous individuals from Pod Mrčaru (Figure 2, Table 1). *Rickettsiella* was more abundant in all individuals from the three populations sampled in 2016 compared to 2014 (Figure 2). *Bacillus* and *Desulfovibrio* were more abundant collectively in insular populations (Figure 2). No sex differences were detected (Table 1). Inter-individual variation in the gut microbiota was, however, present (see Appendix A). There are other important taxa that vary, although not statistically significantly so, amongst the insectivorous and omnivorous lizards including Peptostreptococcaceae (more abundant in omnivorous Pod Mrčaru lizards) and *Akkermansia* (more abundant in insectivorous lizards; Figure 3; Appendix A).

### 3.2. Alpha Diversity Analyses

No significant model of alpha-diversity differences was retained for sex, location, insularity, and year of sampling. Only the diet model was significant (Table 2; χ^2^ = 5.18, *p* = 0.023). The microbial diversity of omnivorous lizards was higher than the microbial diversity of insectivorous lizards (Appendix A).

### 3.3. Beta Diversity Analyses

Only lizard population of origin (location) was significant for all the distance models. Location explained 9% of the variance for unweighted UniFrac distances, while it explained 7% of the variance for weighted UniFrac distances, yet only 5% of the variance for Bray–Curtis distances (Table 3). This trend is observable in the PCoA where the principal axes present between 14% and 16% of the variation (Figure 4). Bray–Curtis and unweighted UniFrac data dispersion were different between omnivorous and insectivorous lizard gut microbiota (Table 4). Gut microbiotas of insectivorous lizards were more dissimilar (with an average distance to the median of 0.35) than gut microbiotas of the omnivorous lizards (average distance to median of 0.31). Dispersion is also different between sampling years for Bray–Curtis and for weighted UniFrac (Table 4). The 2014 samples were more similar (with an average distance to median of 0.23) than the 2016 samples (with an average distance to median of 0.29).

Because we studied insectivorous lizards from both the mainland and from the island of Pod Kopište, we used db-RDA on a subset of the OTUs of these insectivorous populations to disentangle the effect of insularity. Population of origin (location) was the only significant variable in Bray–Curtis and weighted UniFrac distances and explained 3.4% of the variation (Table 5). This factor was also significant in unweighted UniFrac analyses, explaining 5.8% of the variation in unweighted Unifrac distances, with year and sex parsed out. Year and sex explained less variation (respectively 1.5% and 2.3%) when the two other variables are taken into account. Altogether, the unweighted UniFrac model explained 8% of the variation (Table 5). The PCoA shows the predominant influence of the population of origin with the principal axis presenting 16% to 18% of variation (Figure 4D–F). We then performed PERMDISP tests to detect group homogeneity according to the explanatory variables. There is only a difference in dispersion between males and females in insectivorous lizard gut microbiota based on Bray–Curtis and weighted UniFrac distances (Table 6).

## 4. Discussion

We found support for each of our hypotheses, as specific microbial taxa in the lizards’ guts varied by diet, others with location, and some with time (Figure 2). Overall, we did not see large-scale shifts in the entire microbiota (Appendix A), as one may expect given that we investigated population-level differences (Appendix A). The morphology of the Pod Mrčaru *P. siculus* lizards changed rapidly (~30 years) in line with a dietary shift towards the consumption of more plant material (higher bite force, larger body size, evolution of caecal valves, longer guts; [17]). Moreover, the Pod Mrčaru *P. siculus* digest plant material more efficiently (by about 10%) than do the Pod Kopište lizards [19]. Consistent with most of the digestive physiological differences among the Pod Mrčaru and Pod Kopište *P. siculus* being isolated to the hindgut [19], we observed differences in the hindgut microbiota among these lizard populations, and diet was a significant indicator of microbiota diversity (Table 2). The microbial diversity of omnivorous Pod Mrčaru lizards was higher than the microbial diversity of insectivorous lizards (Appendix A), and the Archaean, *Methanobrevibacter*, was significantly more abundant in omnivorous individuals, with diet being a major factor in their abundance (Table 1). Although one species of *Methanobrevibacter* (*M. smithii*) is commonly found in the human gut microbiome [59], this genus is also associated with shifts in microbiome function. For instance, *Methanobrevibacter* is methanogenic and leads to more polysaccharide degradation by bacterial species, greater levels of microbial fermentation, and is often associated with obesity and type two diabetes in rodent models [59,60]. Perhaps this taxon is aiding the Pod Mrčaru lizards in acquiring sufficient energy from their plant-rich diet. One caveat here is that, based on the *Methanobrevibacter* abundance, one would predict greater levels of microbial fermentation in the hindguts of the Pod Mrčaru lizards than in the insectivorous population from Pod Kopište, but in fact the opposite was found [19]. Thus, *Methanobrevibacter* may play other roles in the digestive process than just the production of short chain fatty acids in *P. siculus* [4]. More Pod Mrčaru *P. siculus* should be screened for short chain fatty acid concentrations in their hindguts to determine whether fermentation matters for these lizards or not.

Bacteria in the family Peptostreptococcaceae were more abundant in the Pod Mrčaru omnivorous lizards (Table 2; Figure 3). Although the detailed function of these taxa is unknown, they have been associated with lower protein diets in porcine models [61], and protein digestion more broadly [62], suggesting that they could aid in amino acid scavenging. A previous study has shown that trypsin activity is higher in the hindguts of the Pod Mrčaru lizards, suggesting that protein scavenging is something that occurs in these animals [19]. Interestingly, the genus *Helicobacter* was abundant in the Pod Mrčaru lizards (>2.5% relative abundance), whereas this genus composed less than 0.2% of the reads in the insectivorous populations (Figure 3; Appendix A). Several *Helicobacter* taxa are known from intestinal environments, including other *Podarcis* species [25] and other lizards [6,24], and they may be able to perform many biochemical functions [63]. What these bacteria can do in these lizards requires further study, however. Other bacteria observed in the omnivorous lizards belong to the genus *Rickettsiella* (Figure 3) and were variable in abundance among years (Table 1; Figure 2). *Rickettsiella* are intracellular parasites in a range of host organisms, including insects [64] and mollusks [65]. It remains, however, unknown what the role of this genus could be in *P. siculus*. It is peculiar that it is only prominent in the omnivorous population, but also varied with time (Figure 2). Finally, some of the microbial abundances in the Pod Mrčaru lizards (enriched in *Methanobrevibacter*, depleted in *Akkermansia*) are consistent with dysbiosis (type two diabetes, obesity) in mammalian models including humans [66]. An altered metabolism, including insulin resistance and dyslipidemia, may be advantageous in nutrient-limited environments as seen in cavefishes [67]. Perhaps something similar is happening the Pod Mrčaru lizards, and this could be the focus of future microbiome-host studies in these lizards.

In the island lizards more generally, but in the Pod Kopište lizards more specifically, a member of the genus *Bacillus* appears to be abundant (Table 1; Figure 2 and Figure 3). With over 266 named species, and many biochemical functions known, species in the genus *Bacillus* can perform multiple functions ranging from enzyme secretion to short chain fatty acid (SCFA) synthesis [68]. The island lizards generally have higher b-glucosidase activities in their guts than mainland lizards [19], and perhaps the *Bacillus* may be the source [68]. *Citrobacter*, which are a member of the Enterobacteriacea, were more abundant in the insectivorous lizards (Figure 3, Appendix A). This organism is known to inhabit intestinal environments, and indeed, a wide range of habitats [69], but their role in the gut also remains to be determined. *Akkermansia* is a common intestinal denizen that is more abundant in Pod Kopište than in Pod Mrčaru lizards (Figure 3). *Akkermansia* live in the mucosal layer of mammalian intestines, digest mucins, and play roles in immune function, as well as mucus and peptide secretion [70,71]. A member of this genus was abundant in all lizards, but more so in the insectivorous populations (Figure 3, Appendix A). Interestingly, *Akkermansia* can become more abundant during starvation in many host taxa [62,72] since they can digest mucus [70], and there is a negative relationship between *Akkermansia* abundance and obesity in rodent models: fewer *Akkermansia* equates with the obese phenotype [71]. With *Methanobrevibacter* abundance positively associated with obesity in mammalian models [59,60], and *Akkermansia* negatively so [71], the Pod Mrčaru lizards display both patterns, again, suggesting that they rely on some level on microbial help to gain sufficient energy from their plant-rich diet. *Desulfovibrio*, sulfur reducing bacteria [73], were also more abundant in the island lizards than in the mainland ones (Table 1, Figure 2). Members of this genus have been shown to be more active when exposed to chitin breakdown products [73] and were found in greater abundance in omnivorous lizards fed more plant material in the laboratory [20]. Although all of the lizard populations ingest chitin (part of the exoskeleton of their insect prey), the Pod Kopište and mainland lizards presumably consume more chitin than the Pod Mrčaru lizards. Nevertheless, a previous study showed that only the Pod Kopište lizards displayed elevated N-acetyl-β-D-glucosaminidase (NAG) activity in their guts, indicating potential elevated breakdown of chitin [19].

Previous studies on highly specialized herbivorous lizards like Galápagos marine and land iguanas found differences in the fecal microbiota of these two lizards with a greater diversity of OTUs in the land iguana [21]. Land iguanas would consume more fibrous material (e.g., cellulose) than the marine iguana, consuming marine algae [74]. A subsequent metagenomic study found that the fecal microbiomes between these two lizards were more similar to each other than either was to mammalian herbivores, and that genes for enzymes that could aid in fiber digestion (e.g., cellulases) were abundant in both species, suggesting a functional role of the microbiome in digestion. Moreover, two omnivorous liolaemid lizard species had microbiomes that were more similar to each other than to a distantly related herbivorous lizard [6]. This example, and the iguana studies [21,22], agree with the tenets of Phylosymbiosis—i.e., that there is a correlation between host identity and microbial community [12]. In this study, we found similar to Baldo et al. (2018) for *P. lilfordi* that the collection locality matters for *P. siculus* (Figure 4) [25]. Location was a significant predictor for *Methanobrevibacter*, *Bacillus*, and *Delsulfovibrio* (Table 1; Figure 2). The microbiotas of the Pod Mrčaru lizards consistently grouped separately from the other populations, and those of the Pod Kopište lizards were separate from those of the mainland lizards from Split (Figure 4). Indeed, the two insectivorous populations showed the most heterogeneity amongst them, depending on the analyses used (Figure 4). The gut microbial diversity of insectivorous lizards overall was more heterogeneous than the gut microbial diversity of omnivorous lizards (Figure 4) because insectivorous lizards living on the continent had different gut microbial communities than those living on Pod Kopište. Likewise, *Bacillus* and *Desulfovibrio* were more abundant in the gut microbiotas in insular lizards (Figure 2). The local environment is likely very important in structuring the microbiota in lizards, as a strong similarity between gut microbiota and those observed on plants eaten by lizards was observed in a previous study [20]. The enteric microbial communities of *P. lilfordi* in Menorca were structured by the age of isolation of the islet and the local environment [25] in accordance with our results. Other potential drivers of differences in gut microbial diversity such as sex or year of sampling were largely non-significant as observed in other studies of lizard gut microbiota [11,75].

The vast majority of the bacteria observed in previous microbiome studies in lizards were also observed in *P. siculus* ([6,20,22,24,25,27,75]; Appendix A). Our results showed moderate but significant differences in the microbiota between omnivorous and insectivorous lizards, and the omnivorous lizards can digest plant material moderately (~10%), but significantly better than lizards from the insectivorous population [19]. Given that other studies have demonstrated changes in microbiota when feeding lizards different diets in captivity [20], this likely does not reflect a time constraint but rather the availability of different microbiota in the immediate environment of the lizards [6], combined with flexibility in gut structure and function [20].

## 5. Conclusions

Our results show that the microbial diversity was greater in omnivorous lizards compared to insectivorous lizards of the same species. Moreover, their gut microbiota was characterized by a significant increase in *Methanobrevibacter*. Insectivorous lizards on the mainland had different gut microbial communities than their insular counterparts on the island of Pod Kopište, with *Bacillus* and *Desulfovibrio* being more abundant in insular lizards. Finally, the population of origin is also an important driver of the composition of the gut microbiota. In summary, both diet and the local environment contribute to differences in gut microbial composition in natural populations of these lizards. Importantly, however, our data show that changes in gut microbiota may take place on ecological timescales. Further studies investigating the functional diversity of the gut microbiota and digestive capabilities of the omnivorous lizards (fermentation in particular) might shed light on the role of the microbiota in allowing these lizards to thrive on a plant-rich diet.

## Figures and Tables

**Figure 1 microorganisms-10-01550-f001:**
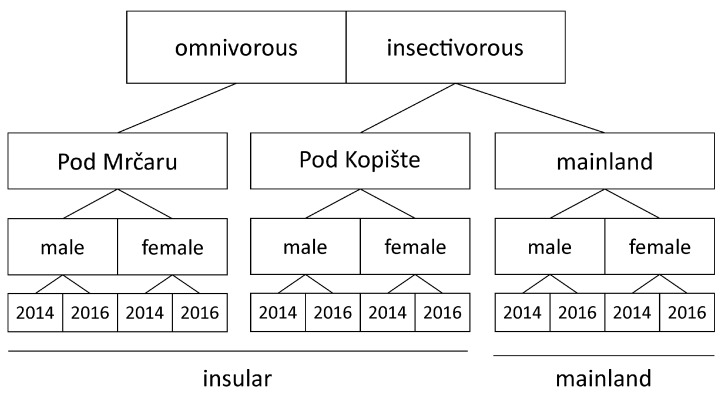
Schematic drawing showing the sampling locations, the diet, and the overall sampling and testing strategy.

**Figure 2 microorganisms-10-01550-f002:**
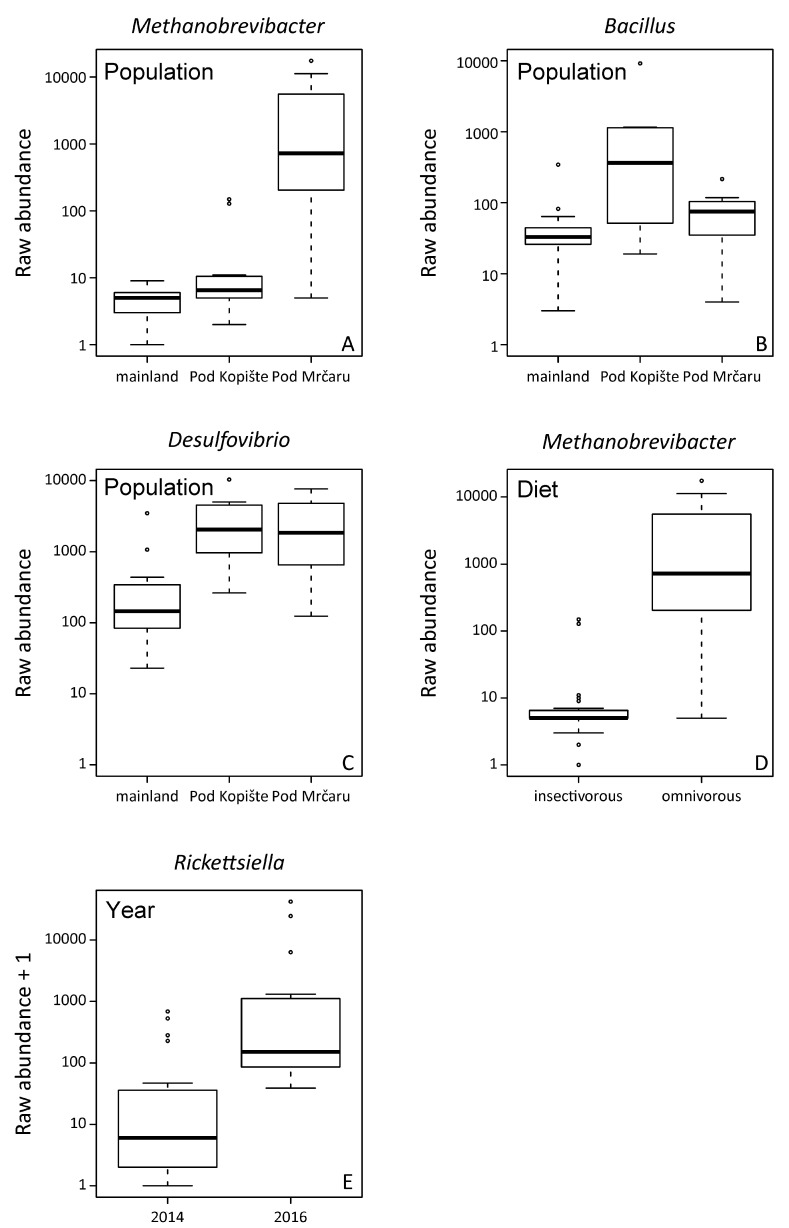
Significantly different abundant taxa detected with Analysis of Composition of Microbiomes (ANCOM) The analysis was performed on unrarefied OTUs with a relative abundance higher or equal to 1% across the whole data set. Significant taxa at *p* < 0.05 after Benjamini–Hochberg correction. (**A**) Differences in the abundance of *Methanobrevibacter* between populations, with lizards from Pod Mrčaru showing higher abundance; (**B**) detected differences across populations in the abundance of *Bacillus*, with a higher abundance in the lizards from Pod Kopište; (**C**) detected differences across populations in the abundance of *Desulfovibrio* showing a higher abundance in lizards from both islands; (**D**) differences between dietary groups in the abundance of *Methanobrevibacter*, being more abundant in the omnivorous lizards; (**E**) differences between the year of sampling in *Rickettsiella* with 2016 showing a higher abundance.

**Figure 3 microorganisms-10-01550-f003:**
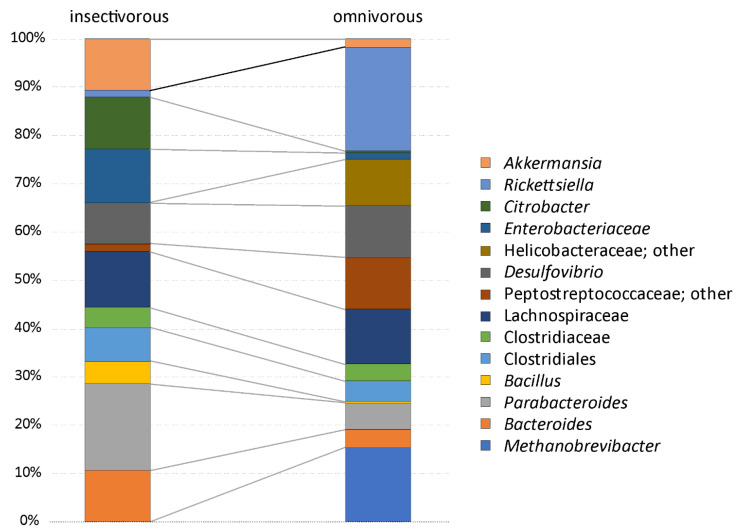
Mean relative abundance composition of lizard gut microbiota in lizards with a different diet. The analysis was performed on the unrarefied OTU table with OTUs, showing a relative abundance higher than or equal to 1%.

**Figure 4 microorganisms-10-01550-f004:**
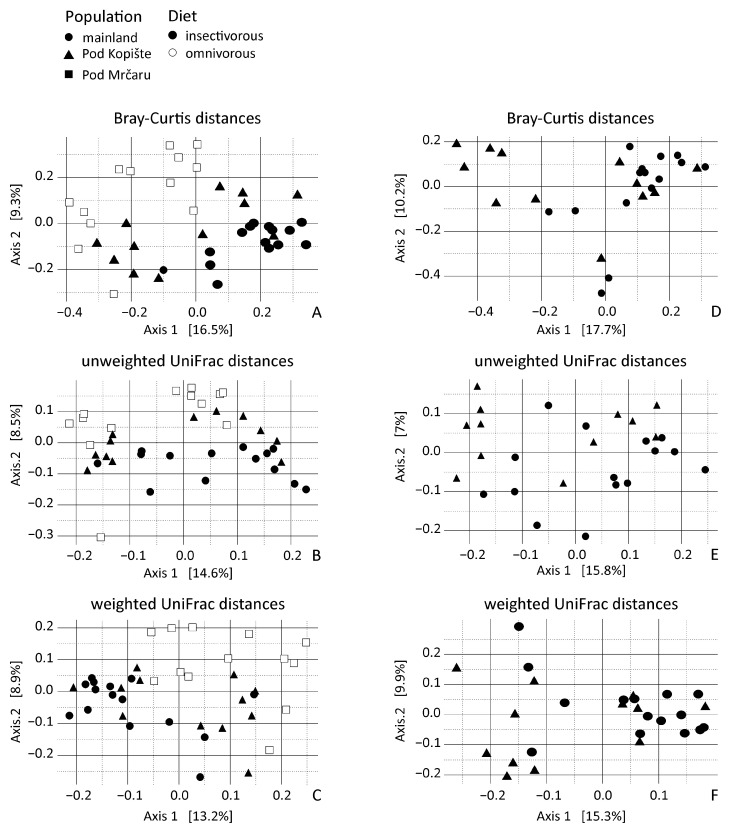
Principal coordinate analyses of rarefied lizard gut microbiota beta diversity from individuals with different diets and from different localities. (**A**) principal coordinate analysis of Bray–Curtis distance across all three populations; (**B**) principal coordinate analysis of unweighted UniFrac distances across all three populations; (**C**) Principal coordinate analysis of weighted UniFrac distances across all three populations; (**D**) principal coordinate analysis of Bray–Curtis distances including only lizards from the mainland and Pod Kopište; (**E**) Principal coordinate analysis of unweighted UniFrac distances including only lizards from the mainland and Pod Kopište; (**F**) principal coordinate analysis of weighted UniFrac distances including only lizards from the mainland and Pod Kopište. Each point represents a sample from an individual lizard. Circles represent lizards from the mainland (Split), triangles represent lizards from Pod Kopište, and squares represent lizards from Pod Mrčaru. Open symbols represent omnivorous lizards and filled symbols represent insectivorous lizards.

**Table 1 microorganisms-10-01550-t001:** Statistic Analysis of Composition of Microbiomes (ANCOM) on most abundant taxa.

	W-Statistic			
	Diet	Location	Sex	Year
*Methanobrevibacter*	13 *	13 *	0	3
Bacteroides	4	7	0	7
Parabacteroides	6	8	0	9
Bacillus	6	11 *	0	6
Clostridiales	8	8	0	8
Clostridiaceae	7	8	0	6
Lachnospiraceae	7	7	0	5
Peptostreptococcaceae; Other	8	10	0	6
*Desulfovibrio*	7	11 *	0	6
Helicobacteraceae; Other	8	10	0	5
Enterobacteriaceae	7	8	0	3
*Citrobacter*	7	8	0	5
*Rickettsiella*	4	10	0	13 *
*Akkermansia*	8	9	0	4

Table entries are W-statistic values. * indicates significance at *p* < 0.05 after Benjamini–Hochberg correction.

**Table 2 microorganisms-10-01550-t002:** Linear mixed model of Shannon exponential (true) diversity tested by Likelihood Ratio Test with chi-square distribution.

Fixed Effect	Random Effect	Likelihood Ratio	*p*
Diet	Sex, Year, Insularity, Location	5.18	**0.023**
Location	Sex, Year, Insularity, Diet	4.25	0.12
Insularity	Sex, Year, Location, Diet	0.24	0.62
Sex	Year, Insularity, Location, Diet	0.45	0.50
Year	Sex, Insularity, Location, Diet	2.38	0.12

Bold indicate significant *p*.

**Table 3 microorganisms-10-01550-t003:** db-RDA of Bray–Curtis, unweighted and weighted UniFrac distances of gut microbiota of significant variables selected with forward selection.

Distance Measure	Model	Db-RDA		
		*F*	*R*^2^ Adjusted	*p*
Bray–Curtis	Location	2.02	0.050	<0.001
Unweighted UniFrac	Location	2.86	0.087	<0.001
Weighted UniFrac	Location	2.33	0.066	0.001

Inertia of response variable matrix is 11.43 for Bray–Curtis db-RDA (14.82% constrained and 85.18% unconstrained, 4.20 for unweighted UniFrac db-RDA (18.52% constrained and 81.48% unconstrained) and 3.03 for weighted UniFrac db-RDA (16.44% constrained and 85.38% unconstrained).

**Table 4 microorganisms-10-01550-t004:** Analysis of lizard gut microbiota group homogeneity (PERMDISP) by factors.

	Factor	*F*	*p*
Bray–Curtis	**Diet**	5.49	**0.024**
	Location	1.12	0.35
	Insularity	0.79	0.38
	**Year**	4.43	**0.041**
	Sex	2.83	0.099
Unweighted UniFrac	**Diet**	4.32	**0.032**
	Location	1.21	0.34
	Insularity	0.39	0.59
	Year	0.37	0.57
	Sex	3.07	0.09
Weighted UniFrac	Diet	0.11	0.74
	Location	0.02	0.98
	Insularity	0.53	0.47
	**Year**	11.11	**0.0014**
	Sex	1.37	0.24

Factors and *p* highlighted in bold showed significant differences.

**Table 5 microorganisms-10-01550-t005:** db-RDA of Bray–Curtis, unweighted, and weighted UniFrac distances of gut microbiota of significant variables selected with forward selection for insectivorous lizards only.

Distance Measure	Model	Controlled Factors	*F*	*R*^2^ Adjusted	*p*
Bray–Curtis	Location	NA	1.96	0.034	0.006
Unweighted UniFrac	Location + Sex + Year	NA	1.72	0.077	<0.001
	Location	Sex + Year	2.15	0.058	0.0036
	Sex	Location + Year	1.61	0.023	0.0086
	Year	Location + Sex	1.38	0.015	0.028
Weighted UniFrac	Location	NA	1.96	0.036	0.0079

Total inertia of response variable matrix is 7.35 for Bray–Curtis db-RDA (14.82% constrained and 85.18% unconstrained), 3.21 for unweighted UniFrac db-RDA (18.31% constrained and 81.69% unconstrained), and 1.76 for weighted UniFrac db-RDA (17.34% constrained and 83.29% unconstrained).

**Table 6 microorganisms-10-01550-t006:** Analysis of insectivorous lizard gut microbiota group homogeneity (PERMDISP) by factor.

	Factor	*F*	*p*
Bray–Curtis	Location	3.09	0.10
	Year	3.09	0.088
	**Sex**	4.93	0.034
Unweighted UniFrac	Location	0.89	0.35
	Year	0.61	0.44
	**Sex**	6.76	0.014
Weighted UniFrac	Location	0.78	0.39
	Year	0.389	0.55
	Sex	0.82	0.38

Factors highlighted in bold showed significant differences.

## Data Availability

The data presented in this study are openly available in FigShare at doi:10.6084/m9.figshare.20393277.

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
