# Peer review of "Proximate Drivers of Population-Level Lizard Gut Microbial Diversity: Impacts of Diet, Insularity, and Local Environment"

_microorganisms, 2022, doi:10.3390/microorganisms10081550_

Round 1

Reviewer 1 Report

Abstract must be described in different ways (for example first aim of study, methods, results and conclusion).

The end of the introduction is aim missing.

Gut samples are not described very well and need some correction.

Statistically analyses must be better described.

Results and discussion are very well described.

Conclusion with most important data is missing.

Author Response

Referee 1

The end of the introduction is aim missing

OUR REPLY: We now end the introduction by stating the aim of the study as suggested.

Gut samples are not described very well and need some correction.

OUR REPLY: we carefully double checked the description and Table S1 which lists all samples.

Statistically analyses must be better described.

OUR REPLY: we tried to clarify the description of the statistics.

Results and discussion are very well described

OUR REPLY: we thank the reviewer for this positive note.

Conclusion with most important data is missing.

OUR REPLY: We added a conclusion as suggested.

We attached the new version of the manuscript.

Reviewer 2 Report

Many thanks to the authors for their efforts on this manuscript, but I have a few notes:

-       The author stated that 40 samples were collected in 2014 and 2016. In line 174, 38 samples are mentioned?

-       Lines 137, 138: ‘A.H., pers. obs.’ What do these abbreviations mean?

-       Lines 140, 141: ‘(Anamaria Štambuk, pers.). Is this a reference? If so, complete its data and add it to the reference list

-       Line 265: correct ‘Bacilus’ to ‘Bacillus’

-       In lines 285 - 322 you referred to Tables 3, 4, 5 and 6, but they are missing and also not found in the supplementary data.

-       References:

-       There is an errors in the way references are written in the text, for example in line 86; ‘Hong, Wheeler, Cann & Mackie, 2011’ , you should write ‘Hong et al., 2011’

-       Some references like ‘Baldo’ are written on line 427 “e Baldo, Riera, Mitsi, and Pretus (2018)”. it is written in line 97 “Baldo et al., 2018”.

-       Complete the reference No. 49 in the list

-       Reference No. 56 is written in the text for the year 2013 but in the list for the year 2014, check it out

-       References No. 71 and 72 refer to Vervust et al., 2009. check it out

-       Reference No. 73 has been quoted 13 times. It is better to avoid this repetition and to cite other references

Author Response

Referee 2

- The author stated that 40 samples were collected in and 2016. In line 174, 38 samples are mentioned?

OUR REPLY: this was an error; supplementary table 1 lists 40 samples.

- Lines 137, 138: ‘A.H. pers. obs.’ What do these abbreviations mean.

OUR REPLY: this refers to Anthony Herrel (senior author of the paper), personal observation. This is commonly abbreviated as such but if the editor prefers then we can modify this as stated.

- Lines 140, 141: ‘Anamaria Stambuk, pers.). Is this a reference? If so, complete its data and add it to the references list.

OUR REPLY: in the text is indicated ‘Anamaria Štambuk, pers. comm.) which refers to a personal communication of our colleague Anamaria with whom we collaborate on a genomic study on the host lizards. This is also a pretty common way to refer to personal communications of unpublished data. Again, we would be very willing to put this in full if the editor requires us to do so.

- Line 265: correct Bacilus to Bacillus.

OUR REPLY: our apologies, this has been corrected.

- In lines 285-322 you referred to Tables 3,4,5 and 6 but they are missing and also not found in the Supplementary data.

OUR REPLY: We are very sorry but these tables exist and should have been uploaded. We have no idea what went wrong. We will double-check upon resubmission to assure these are present.

References:

- There is an error in the way references are written in the text, for example in line 86; ‘Hong, Wheeler, Cann & Mackie, 2011’, you should write ‘Hong et al., 2011’

OUR REPLY: this has been corrected.

- Some references like ‘Baldo’ are written on line 427 “e Baldo, Riera, Mitsi, and Pretus (2018)”. it is written in line 97 “Baldo et al., 2018”.

OUR REPLY: this has been corrected.

- Complete the reference No. 49 in the list.

OUR REPLY: this is a reference to an R package and as far as we know this is the correct reference.

- Reference No. 56 is written in the text for the year 2013 but in the list for the year 2014, check it out.

OUR REPLY: Thanks for catching this, it has been corrected.

- References No. 71 and 72 refer to Vervust et al., 2009. check it out.

These are two different papers referring to different topics. We have added a and b to make the distinction clearer.

- Reference No. 73 has been quoted 13 times. It is better to avoid this repetition and to cite other references.

OUR REPLY: we have reduced the number of citations, but the paper is highly relevant as it is a physiological study on the same populations.

Reviewer 3 Report

The authors herein presented an interesting paper on microbiota diversity related to diet and insularity. Experminal section Is well described, resulta are nicely organized and well fit the conclusion. Additionally also the referente appear appropriate.

Author Response

Referee 3

The authors herein presented an interesting paper on microbiota diversity related to diet and insularity. Experimental section is well described, results are nicely organized and well fit the conclusion. Additionally, the references appear appropriate.

OUR REPLY: we that the reviewer for her/his positive evaluation of our manuscript.

Round 2

Reviewer 1 Report

In the future authors need all changes marked in manuscript with different color or tracking changes.

Conclusion is part where authors described important results and it is a separate chapter. Conclusion must be redescribed without references.

Author Response

- Conclusion is part where authors described important results and it is a separate chapter. Conclusion must be redescribed without references.

- Thank you for this comment. We created a separate section containing the conclusion.